# 3dDNA: A Computational Method of Building DNA 3D Structures

**DOI:** 10.3390/molecules27185936

**Published:** 2022-09-13

**Authors:** Yi Zhang, Yiduo Xiong, Yi Xiao

**Affiliations:** School of Physics and Key Laboratory of Molecular Biophysics of the Ministry of Education, Huazhong University of Science and Technology, Wuhan 430074, China

**Keywords:** 3dDNA, DNA, 3D template libraries, 3D structure prediction

## Abstract

Considerable progress has been made in the prediction methods of 3D structures of RNAs. In contrast, no such methods are available for DNAs. The determination of 3D structures of the latter is also increasingly needed for understanding their functions and designing new DNA molecules. Since the number of experimental structures of DNA is limited at present, here, we propose a computational and template-based method, 3dDNA, which combines DNA and RNA template libraries to predict DNA 3D structures. It was benchmarked on three test sets with different numbers of chains, and the results show that 3dDNA can predict DNA 3D structures with a mean RMSD of about 2.36 Å for those with one or two chains and fewer than 4 Å with three or more chains.

## 1. Introduction

There is an increasing need for determining 3D structures of DNA. A typical ex-ample is DNA aptamer selection [1,2,3]. Aptamers for a desired target are selected from a large oligonucleotide library, usually through SELEX (Sequential Evolution of Ligands by Exponential Enrichment). After multiple rounds of SELEX, the number of resulting aptamer candidates is still very large. Therefore, it is still hard work to deter-mine the best one from the candidates. If we can computationally build 3D structures of the candidates, it could be very helpful for increasing the efficiency of aptamer selection. However, no direct methods of predicting 3D structures of DNAs are currently available, although those for RNAs have been developed for a long time [4,5,6,7,8,9,10,11,12,13,14,15,16,17,18,19,20,21,22,23,24,25,26,27,28,29,30]. In this work we report a prediction method of 3D structures of DNA by extending that of RNA.

Similar to Anfinsen’s assumption for protein [31], RNA 3D structures can be as-sumed to be in the minimum free-energy state, and so their prediction generally includes two steps: sampling conformation space and picking out the minimum free-energy model. Current prediction methods of RNA 3D structures can be divided into two classes roughly: the ab initio approach and template-based approach. The former looks for the 3D structure of an RNA by using molecular dynamics to simulate its folding process, but usually using a coarse-grained model [32] and a reduced force field [33] due to the limitation of computational capacity, which usually makes their accuracies decrease with the increase in RNA length. The latter looks for the 3D structure of an RNA by searching and assembling the 3D templates from experimental RNA structures that have similar sequence or sequence fragments with those of the target RNA. This approach can increase the efficiency of the sampling of conformation space and prediction accuracy and is widely used by many prediction methods of RNA 3D structures, such as ASSEMBLE [5], RNAComposer [6,7,8], and 3dRNA. So, we use the template-based approach for 3dDNA, especially based on 3dRNA, proposed in our laboratory, which can automatically predict the 3D structure of an RNA by assembling 3D templates of Smallest Secondary Elements (SSEs), including stem, hairpin loop, bulge loop, internal loop, open loop, and junction.

In contrast to RNA, there were only indirect methods to predict DNA 3D structure [34,35], which first predicted the 3D structure of the corresponding RNA [5,6], then converted it into that of DNA by replacing the nucleotide U with T, and finally refined the resulting 3D structures through energy minimization. This approach is mainly limited to predict the 3D structures of stem-loop aptamers with smaller loops (six or fewer nucleotides) and the accuracy is more than 4.0 Å of the RMSD. Here, we present a template-based method of building 3D structures of DNA directly. 

## 2. Results

For a target DNA, 3dDNA can give assembled, optimized, and best structures. The assembled structure is one just assembled by using the 3D templates for each SSE of the target DNA and minimized by Amber to avoid atom clash. It can be further optimized by SAMC to give optimized structures. If the perfect template for each SSE of the DNA can be found in the template library, the assembled structure is considered as the best structure, otherwise, the optimized structure is considered as the best structure. During the test, the secondary structure of the target DNA is obtained by X3DNA from its PDB structure. Furthermore, we removed 3D templates extracted from the 3D structures of the DNAs in the test set from the SSE 3D templates library. 

### 2.1. Prediction Accuracy of DNAs with Single Chain

To benchmark 3dDNA, all the sequences with single chain were clustered using CD-HIT-EST [36] with the cutoff of 80% firstly, then 31 DNAs were selected as Test Set 1 according to the criteria: length and 2D structure. The detailed information and predictions of this test set are given in Figure 1 and Table 1. When considering the open loops, the mean RMSD values of the assembled, optimized, and best structures are 3.29 Å, 3.34 Å, and 2.58 Å, respectively. If not considering the open loops, they are 2.28 Å, 3.08 Å, and 2.36 Å. In fact, we found that the structures of the open loops in DNAs are very flexible, which will affect the reasonable evaluation of the prediction performance of 3dDNA. As shown in Figure 2, among the 20 native models measured by NMR experiment, the open-loop structure of a DNA (PDB ID 1EN1) is very flexible. When the open loop is not considered, the RMSD of the assembled structure of 1EN1 is reduced from 8.32 Å to 4.2 Å. 

It is worthwhile to note that the assembled structures of only 9 of 31 DNAs have RMSDs higher than the mean one. Among them, DNA 1OMH without perfect templates can be optimized more closely to the native structure, while DNA 2FDC_1, 2VIC_2, and 1EZN with perfect templates generally show poor effect of optimization; Figure 3 shows two detailed examples. These results show that the best structure of a DNA can represent its native structure in most cases. 

### 2.2. Prediction Accuracy of DNAs with Double Chains

The DNAs with a single chain are only a small part of DNAs, 87% of which have two chains. A total of 56 DNAs were selected as Test Set 2 according to their length and 2D structure complexity. The detailed information of DNAs of this test set is given in Table 2. The mean RMSD value of the assembled, optimized structure, and best structures are 3.0 Å, 5.64 Å, and 2.83 Å, respectively (Figure 4 and Table 2). Overall, 3dDNA can reach higher prediction accuracy (2.83 Å on average) for the best structure of double-chain DNA predictions with perfect templates, with only 11 cases of high RMSDs (>4.0 Å) out of 56 cases. As shown in Figure 5, the assembled structure of 3RB6_1 with perfect template becomes much worse when optimized, and so it is the ideal predicted structure. However, for the DNA 4DAV without perfect templates, the assembled structure can be further optimized, and the optimized structure is the ideal predicted structure.

### 2.3. Prediction Accuracy of DNAs with Multiple Chains

To further test the prediction accuracy of 3dDNA, we also built a test set (Test Set 3) of 29 DNAs selected from 236 DNAs with multi-chains according to the following standard: length, 2D structure complexity, and number of chains. As shown in Figure 6 and Table 3, the mean RMSDs of the assembled, optimized, and best structures are 6.9 Å, 7.64 Å, and 5.28 Å, respectively. Similar to the DNAs with single and double chains, 3dDNA can give near-native predictions for 3D structures of the DNAs with multi-chains when having perfect templates in 3dDNA_Lib, and the assembled structures of the multi-chain DNAs without perfect templates can be further optimized. As shown in Figure 7, the assembled structure of 1TW8 with perfect template is more accurate than the optimized structure. There are also some exceptions, for example, the assembled structure of the DNA 6L74 with imperfect templates is optimized to be bad, which leads to the best structures not being the ideal predicted structures. On average, the prediction accuracy of 3dDNA for DNAs with multi-chains is lower than that for DNAs with single and double chains. In fact, there are a large number of broken loop structures in double-chain and multi-chain DNAs, especially in the latter, but such broken loop structures are very uncommon in the 3D template library, that is, perfect templates are basically not found in the template library, which leads to their prediction accuracy being worse than that of single-chain DNAs.

### 2.4. Comparison with Indirect Method

The 3D structures of some short hairpin aptamers were predicted using the indirect approach mentioned above [34,35]. For comparison, the hairpin aptamers used in the indirect predictions by Jeddi are taken as Test Set 4, which contains 24 small hairpin aptamers with lengths from 7nt to 27nt. The detailed information of the Test Set 4 is given in Table 4. 

Figure 8 and Table 4 show the comparison of 3dDNA and indirect predictions. The mean RMSD values of 3dDNA predictions for assembled, optimized, and best structures are 2.67Å, 3.00Å, and 2.69Å, respectively, and they are significantly smaller than indirect predictions (4Å on average). For the best structures, 3dDNA gives smaller RMSDs for 17 out of 24 DNAs than the indirect predictions. It is noted that all DNAs in Test Set 4 can find the perfect template in the template library, except 1EN1 (the assembled structure with a RMSD of 8.32Å). These results show that the prediction accuracies of 3dDNA are much better than indirect predictions. 

## 3. Method

### 3.1. Classification of DNA Structures

We analyzed all of the DNA 3D structures in PDB and found that they can be divided into 5 classes:

(1) DNA without base pairs: unStru-DNA. The total number is 645, and the proportion of this class is about 12%.

(2) DNA with pure duplex structure, and the base pairs are canonical ones: Helix-DNA. The total number is 977, and the proportion of this class is about 18 %.

(3) DNA with both duplex and loop structures: D-DNA. The total number of the D-DNAs is 3604, and the proportion of this class is about 61%. Of them, the molecules with single, double, and triple chains account for 6%, 87%, and 5%, respectively. A very small part of this class contains more than three chains.

(4) The DNAs containing triple helices: T-DNA. The total number is 134, and the proportion of this class is about 2%.

(5) The DNAs containing quadruple helices or consecutive stacking G-quadruples: G-DNA. The total number is 185, and the proportion of this class is about 3%.

This work only considers the DNAs of the third class, which account for 61% of all DNAs as shown in Figure 9. The DNAs of the first two classes are not considered because one has no stable structures, and one has the standard B-helix structure, while the last two classes are also not considered due to their small number.

### 3.2. Smallest Secondary Elements

3dDNA is extended from our 3dRNA [9,10,11], which is a template-based method of building 3D structures of RNAs by assembling 3D templates of Smallest Secondary Elements (SSEs). The 2D or 3D structures of an RNA or DNA can be decomposed into different types of SSEs. The SSEs are defined as stems, hairpin loops, bulge loops, internal loops, open loops, break loops, and junction loops (multi-branch loops) with connected 2-base-pairs at each end. As an example, Figure 10 shows the process of decomposing DNA 1QSK into five SSEs, which contain two stems or helices, one bulge loop, one open loop, and one break loop. It is noted that any adjoining SSEs have two common base pairs that are superposed when assembling the SSEs into the whole 3D structure. Furthermore, compared with RNA, the break loop is a new type of SSE since most DNAs are multi-chain structures. As shown in Figure 10A, a break loop is marked by a red rectangular box, which means a helix connected to two broken single chains. Although the open loop and the broken loop in Figure 10A look the same, the positions of the helical and loop regions of the two are opposite, resulting in different secondary structures of the two, which directly affect the subsequent template search and module assembly.

### 3.3. DNA SSE 3D Template Library

To build the 3D template library of DNA SSEs, 8460 DNA structures were collected from the RCSB PDB database [39]. Among them, DNAs with fewer than 4 nucleotides or having the same structures as other DNAs were removed first, and then the rest of the DNAs were filtered by “clean”, “mutate”, and “amber”, respectively, where “clean” means to extract only the atoms that contain DNA in the PDB structure, “mutate” means to mutate all the nonstandard bases in DNA as the standard base “AUCG”, and “amber” is mainly used to complete the atomic deletion problem in DNA, referring to the article [30] for these detailed procedures. Furthermore, in the remaining DNAs, those without base pairs, with only pure double helices, or with triplex and quadruple helices were also removed as shown above. Only DNAs with both loop and stem structures were considered. As a result, 3604 DNAs were kept, among which 87% of molecules contain double chains, 6% single chains, 5% triple chains, and 2% four or more chains. Finally, according to the secondary structures [40] of these remaining DNAs, their 3D structures were split into 3D templates according to the SSEs. It is noted that unlike the abundant junctions in RNA, there are only a few 3-way and 4-way junctions in DNA. In this way, all DNA 3D structures were decomposed to form an SSE 3D template library, 3dDNA_Lib, containing 5505 helices, 3949 loops (hairpin loops, bulge loops, internal loops, and junctions), and 3480 break loops. In order to enrich the 3D template library, 3dDNA_Lib and 3dRNA_Lib were combined together to form the DNA template library. If an SSE cannot find a 3D template in 3dDNA_Lib, it can search a template in 3dRNA_Lib with the nucleotide U being replaced by the nucleotide T. The template library 3dRNA_Lib is much larger than 3dDNA_Lib.

### 3.4. The Workflow of 3dDNA

The workflow of 3dDNA is similar to 3dRNA (Figure 11). For a target DNA, its sequence and secondary structure are taken as inputs. According to the secondary structure, the DNA is firstly decomposed into SSEs. Secondly, 3dDNA finds a 3D template for each SSE according to certain rules, with the following priority order: secondary structure topology and sequence similarity. Ideally, more than one template will be found for each SSE in the DNA SSE 3D template library (3dDNA_Lib), and then the template with the highest score will be selected. The scoring of each SSE is defined as follows. Firstly, if the secondary structure is the same, give 5 points, otherwise, give 0 points. Then, if the sequence is traversed, 1 point will be given for the same nucleotide in the loop region and 0.2 points will be given for the same nucleotide in the helix region. If the template of an SSE is not found in 3dDNA_Lib, it will switch to search in the RNA SSE 3D template library (3dRNA_Lib) built in 3dRNA. It may happen that the template of an SSE cannot be found in both template libraries, the bi-residues method or Distance Geometry (DG) algorithm [41] will be called to construct a template for the SSE. Thirdly, we assemble the selected template of each SSE with that of its parent SSE. Any two SSEs are superposed with reference to the two common base pairs according to the Kabsch algorithm [42]. Subsequently, the sequence of the assembled structure is mutated to meet the target sequence, and the assembled models are minimized (1000 steps) with AMBER 98 force field [43,44] to repair the chain connectivity of the assembled structures. In the next step, the templates of all SSEs are analyzed. When all SSEs have perfect templates, the assembled structure is considered as the final structure of the target DNA, otherwise the assembled structure needs to be further optimized. The perfect template means that all SSE in DNA can be found in the DNA template library with matching secondary structures. For an assembled structure that needs to be further optimized, the residue-level simulated annealing Monte Carlo (SAMC) method and a residue-level energy function in 3dRNA are modified by replacing U with T to perform the optimization. The optimized structures are ranked by the residue-level energy function, and the top 5 optimized structures are given as the final output structures. 

The details of the SAMC method and the energy function can be found in our previous work [10]. Briefly, the optimization of an assembled structure uses a coarse-grained model with each residue being represented by 6 atoms: the phosphate atom P of the backbone, C4′ and C2′ atoms from the sugar ring, and C2, C4, and C6 atoms from the base. During the MC process, the smallest movable element is randomly set according to the secondary structure of the initial structure, but the conformations of all helices and short loops (hairpin loops of <5 nt or internal loops of <7 nt) are fixed, except their orientations. In each step of SAMC, we randomly select a moveable element to be translated, rotated around a point, or rotated around an axis. Then, the generated large number of candidate structures are clustered, and the centroid of each cluster is ranked by the coarse-graining model of 3dRNAscore [45], which is a knowledge-based statistical potential that combines distance-dependent energy and torsion-angle-dependent energy. Finally, the ranked top 5 (the default value) optimized DNA are given.

## 4. Conclusions

We developed a template-based method, 3dDNA, for fully automated prediction of the tertiary structures of DNAs from their sequences and secondary structures. Systematic tests show that for sets of DNAs with single chains, two chains, and multi-chains, the prediction accuracy of 3dDNA can reach average RMSDs of 3.13 Å, 2.83 Å, and 5.28 Å, respectively. Therefore, the prediction accuracy of 3D structures of the DNAs with single chains and two chains are similar to that of 3dRNA, but that for multi-chains DNAs is lower than the former and needs to further improve. Furthermore, the accuracy of 3dDNA is significantly higher than the indirect methods. Furthermore, we found that the best structures have lower RMSD values on average than the assembled and optimized structures, the best structure of DNA is the assembled structure if the templates of all SSEs are perfect, otherwise it is the optimized structure. In the future, we hope to include DNAs with triple and quadruple helices [46] in 3dDNA and develop Alphafold-like [47] DNA 3D structure prediction methods. We believe that the increase in RNA and DNA in the experiment will continue to improve the accuracy of 3dDNA.

## Figures and Tables

**Figure 1 molecules-27-05936-f001:**
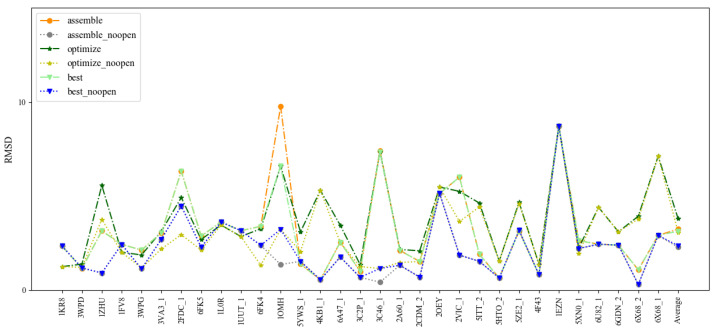
Comparison of the prediction accuracies (all-atom RMSDs) of 3dDNA for the assembled, optimized, and best structures of 31 single-chain DNAs with or without open loops.

**Figure 2 molecules-27-05936-f002:**
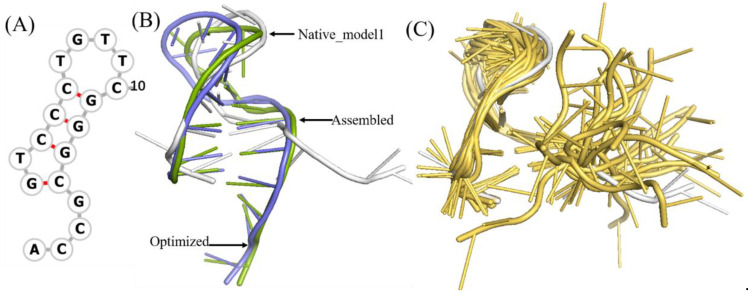
The structural analysis of open loop in a DNA (PDB ID 1EN1). (**A**) The secondary structure of 1EN1. (**B**) The native, assembled model and optimized structures are marked with grey, green, and blue, respectively. The first model in NMR selected as native. When the open loop is not considered, the RMSD of the assembled structure drops from 8.32Å to 4.2Å. (**C**) Twenty models obtained from the NMR experiment, of which model 1 is marked with grey. The 2D and 3D structures are generated by Forna [37] and PyMOL [38], respectively.

**Figure 3 molecules-27-05936-f003:**
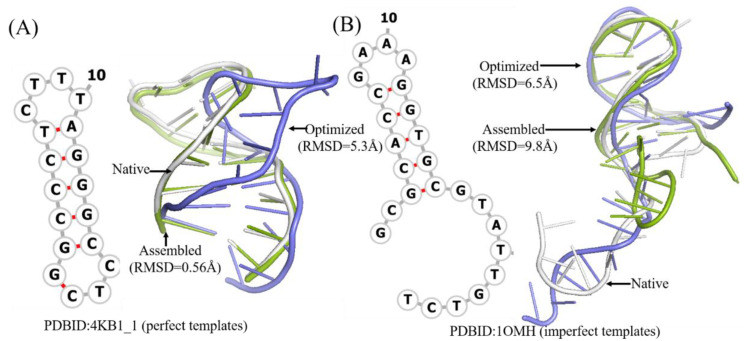
Two example of predicted 3D structures by 3dDNA. (**A**) The 2D and 3D structures of DNA 4KB1_1 with perfect template; the RMSD of assembled (green) and optimized (blue) structures with a native one (grey) are 0.56 Å and 5.3 Å, respectively. (**B**) The 2D and predicted 3D structures of DNA 1OMH with imperfect template, the RMSD of assembled (green) and optimized (blue) structures with a native one (white) are 9.8 Å and 6.5 Å, respectively.

**Figure 4 molecules-27-05936-f004:**
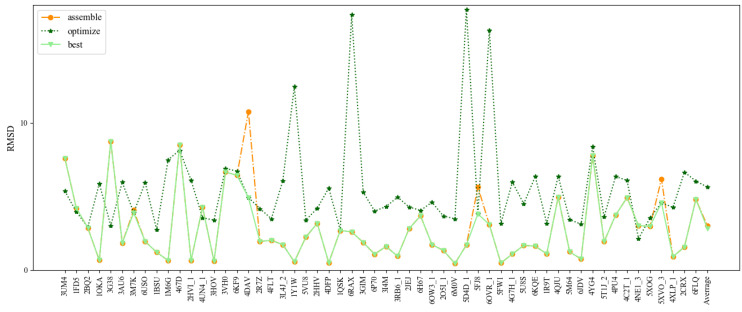
Comparison of the prediction accuracies (all-atom RMSDs) of 3dDNA for the assembled, optimized, and best structure of double-chain DNAs, respectively.

**Figure 5 molecules-27-05936-f005:**
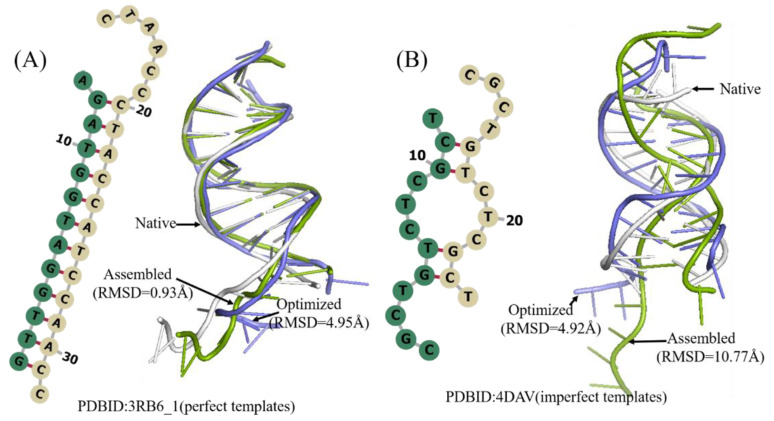
Two examples of the predicted 3D structures of double-chain DNA by 3dDNA. (**A**) The 2D and 3D structures of DNA 3RB6_1 with perfect template, the RMSD of assembled (green) and optimized (blue) structures with a native one (grey) are 0.93 Å and 4.95 Å, respectively. (**B**) The 2D and 3D structures of DNA 4DAV with imperfect template; the RMSD of assembled (green) and optimized (blue) structures with a native one (grey) are 10.77 Å and 4.92 Å, respectively.

**Figure 6 molecules-27-05936-f006:**
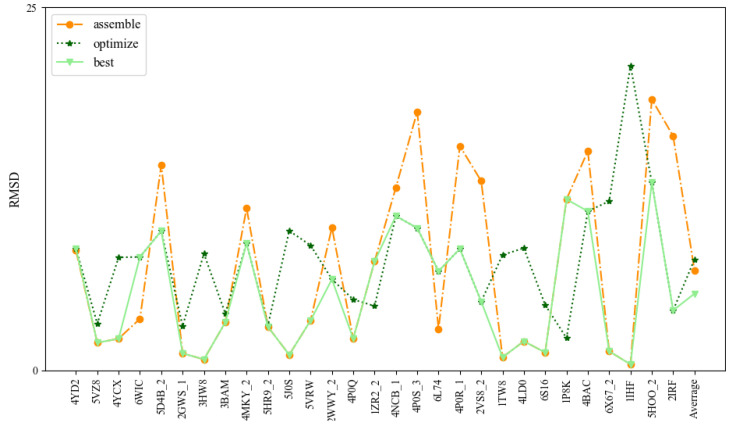
Comparison of the prediction accuracies (all-atom RMSDs) of 3dDNA for assembled, optimized, and best structures of DNAs with multi-chains.

**Figure 7 molecules-27-05936-f007:**
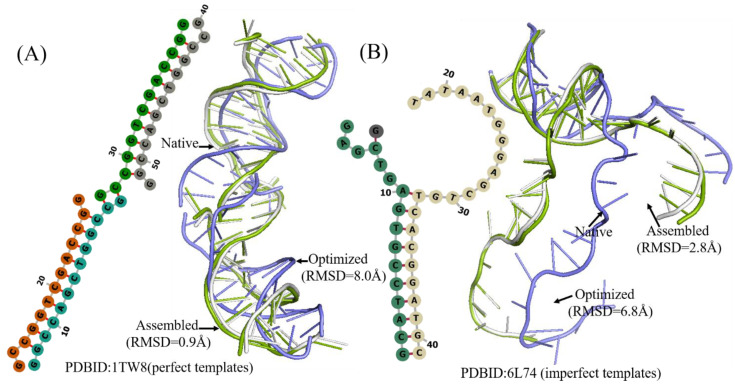
Two examples of the 3D structures of multi-chains DNA predicted by 3dDNA. (**A**) The 2D and 3D structures of DNA 1TW8 with perfect template; the RMSD of assembled (green) and optimized (blue) structures with a native one (grey) are 0.9Å and 8.0 Å, respectively. (**B**) The 2D and 3D structure DNAs 4L74 with imperfect template; the RMSD of assembled (green) and optimized (blue) structures with a native one (grey) are 2.8 Å and 6.8 Å, respectively.

**Figure 8 molecules-27-05936-f008:**
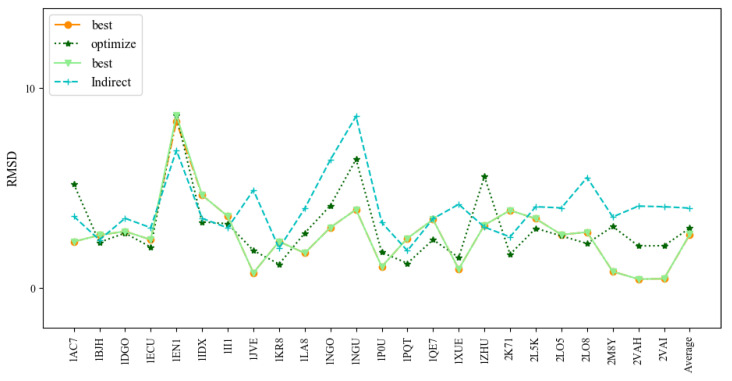
Comparison of the prediction accuracies (all-atom RMSDs) of the assembled, optimized, and best structures of 3dDNA with previous indirect predictions by Jeddi.

**Figure 9 molecules-27-05936-f009:**
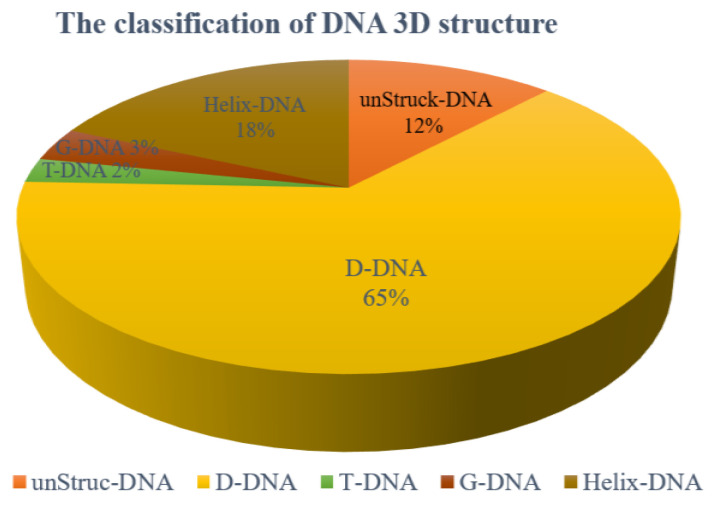
The proportion of five classes of DNA structures in PDB. The 5 classes include unStruck-DNA (DNA without base pairs), Helix-DNA (DNA with pure duplex structure), D-DNA (DNA with both duplex and loop structures), T-DNA (DNAs containing triple helices), and G-DNA (DNAs containing quadruple helices or consecutive stacking G-quadruples).

**Figure 10 molecules-27-05936-f010:**
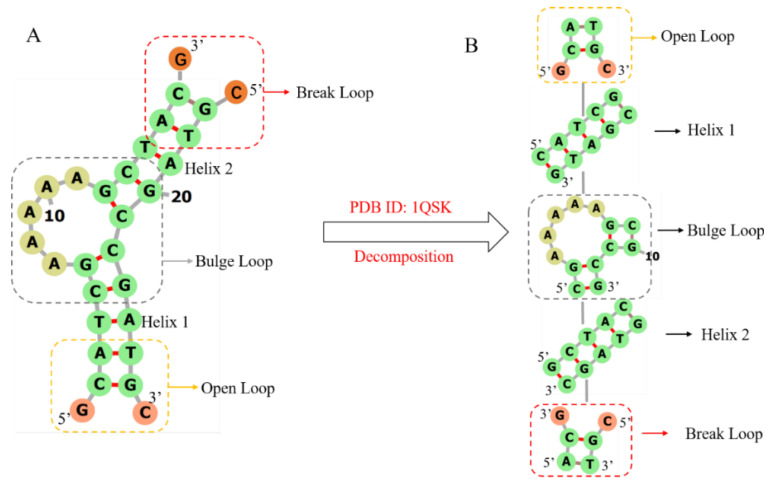
The workflow of decomposing DNA 1QSK to five SSEs. Break loop, bulge loop, and open loop are marked by red, grey, and yellow rectangular boxes, respectively. (**A**) The secondary structure of DNA 1QSK. (**B**) All the SSEs, which together form a secondary structure tree (SST) of DNA 1QSK, including two helices and one open loop with 2D structure “.(()).”, a bulge loop with 2D structure “((…..(())))”, and a break loop with 2D structure “((.&.))”. DNA 2D plots are generated using Forna [37].

**Figure 11 molecules-27-05936-f011:**
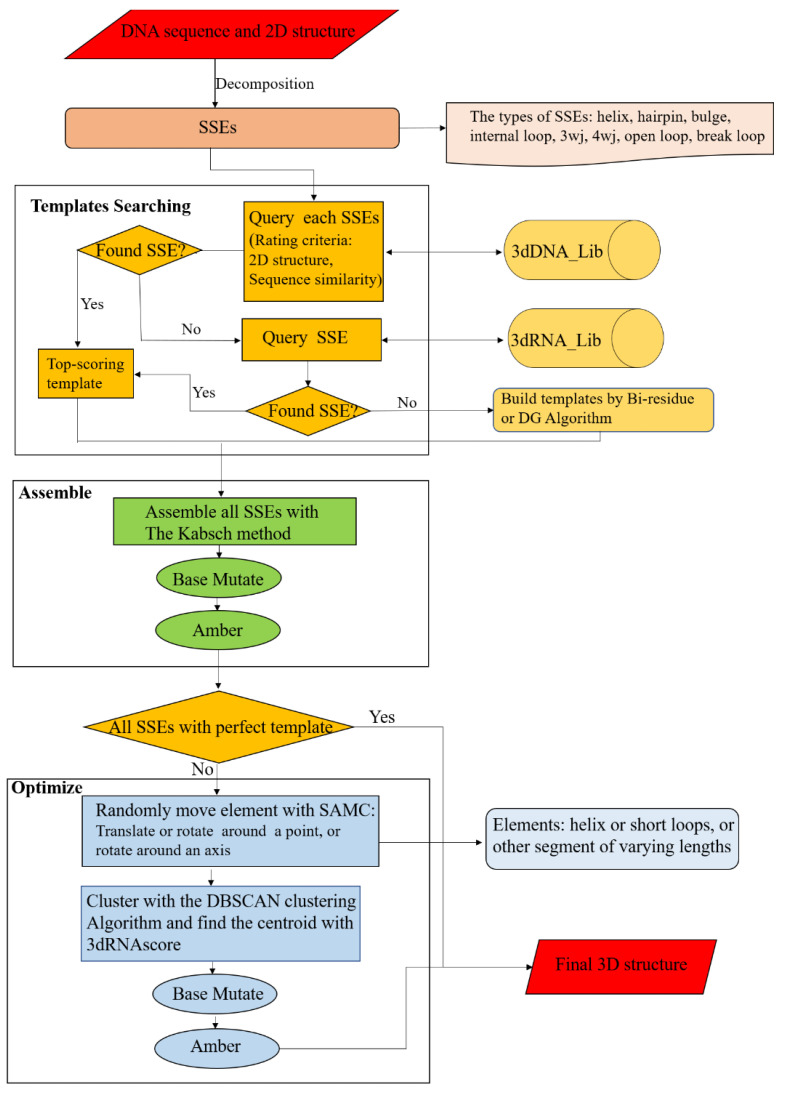
Workflow chart of 3dDNA for DNA 3D structure prediction. Mainly composed of three parts: Templates Searching, Assemble, and Optimization.

**Table 1 molecules-27-05936-t001:** Information and prediction accuracies (RMSD in Å) of 31 single-chain DNAs.

PDB ID	Resolution	Length	2D Structure	Assembled		Optimized		Best	
				All	No Open Loop	All	No Open Loop	All	No Open Loop
1KR8	\	7	((...))	2.34	2.34	1.25	1.25	2.34	2.34
3WPD	2.75	10	((.....)).	1.15	1.18	1.38	1.21	1.15	1.18
1ZHU	\	10	..((...)).	3.15	0.91	5.57	3.76	3.15	0.91
1FV8	\	11	((((…))))	2.43	2.43	2.01	2.01	2.43	2.43
3WPG	2.25	11	((.....))..	2.13	1.13	1.87	1.18	2.13	1.13
3VA3_1	2.71	12	((((..))))..	3.05	2.71	3.13	2.19	3.05	2.71
2FDC_1	3.30	12	(((....)))..	6.32	4.47	4.92	2.95	6.32	4.47
6FK5	2.02	14	(((((...))))).	2.87	2.28	2.69	2.13	2.87	2.28
1L0R	\	14	((…))((...))	3.62	3.62	3.46	3.46	3.62	3.62
1UUT_1	2.00	15	((((((...))))))	3.16	3.16	2.85	2.85	3.16	3.16
6FK4	2.30	16	..(((((...))))).	3.39	2.38	3.26	1.32	3.39	2.38
1OMH	1.95	25	..(((((….))))).........	9.77	1.36	6.59	3.23	6.59	3.23
5YWS_1	2.00	17	..((((((.))))))..	1.38	1.53	3.10	2.06	1.38	1.53
4KB1_1	1.80	18	((((((....)))))..)	0.55	0.55	5.30	5.3	0.55	0.55
6A47_1	1.90	19	..(((((((..))))))).	2.55	1.76	3.45	2.53	2.55	1.76
3C2P_1	2.00	20	.......(((((...)))))	1.00	0.68	1.36	1.25	1.00	0.68
3C46_1	2.00	21	.......(((((...))))).	7.42	0.42	7.36	1.15	7.36	1.15
2A60_1	2.60	22	((((((((....)))).)))).	2.11	1.32	2.16	1.47	2.11	1.32
2CDM_2	2.70	23	(((((....))))).....(..)	1.52	0.69	2.09	1.52	1.52	0.69
2OEY	\	25	(((((..(((((...))))))))))	5.14	5.14	5.48	5.48	5.14	5.14
2VIC_1	2.35	26	....((((((((.....)))).))))	6.00	1.86	5.25	3.65	6.00	1.86
5ITT_2	2.53	26	.(((((.(((((.))))).)))))..	1.91	1.53	4.63	4.42	1.91	1.53
5HTO_2	1.90	30	(((..... [ [..(((((]]...))))))))	0.64	0.64	1.55	1.55	0.64	0.64
5ZE2_1	3.30	31	.((((((((((((......))))))))))))	3.14	3.18	4.67	4.6	3.14	3.18
4F43	2.35	32	(((((((((((((((..)))))))))))))))	0.82	0.82	1.37	1.37	0.82	0.82
1EZN	\	36	(((((((((((..))))((((..))))..)))))))	8.71	8.71	8.73	8.73	8.71	8.71
5XN0_1	2.60	35	..(((((((((((((((...)))))))))))))))	2.63	2.21	2.24	1.95	2.63	2.21
6U82_1	3.21	38	((((((((((((((((...))))))))))...))))))	2.44	2.44	4.40	4.40	2.44	2.44
6GDN_2	3.00	42	((((((((..........(..)..((((..))))))))))))	2.39	2.39	3.1	3.1	2.39	2.39
6X68_2	3.66	50	..(((((((((((((((((((((....)))))))))))))))))))))..	1.08	0.30	3.93	3.78	1.08	0.30
6X68_1	3.66	74	(.((((((((.((((((((((((((((((((((((....)))))))))))))))))))))))).)))))))).)	2.91	2.91	7.12	7.12	2.91	2.91
**Average**	\	\	\	**3.24**	**2.28**	**3.81**	**3.08**	**3.13**	**2.36**

**Table 2 molecules-27-05936-t002:** Information and prediction accuracies (RMSD in Å) of 56 double-chain DNAs.

PDB ID	Resolution	Length	2D Struacture	Assembled	Optimized	Best
3UM4	2.82	9	((..&..))	7.60	5.36	7.60
1FD5	1.10	12	((((.&.)))).	4.19	3.95	4.19
2BQ2	\	13	.((((.&.)))).	2.86	2.91	2.86
1OKA	\	15	(((((….&)))))	0.66	5.87	0.66
3G38	3.04	17	((((..(.&)...))))	8.75	3.02	8.75
3AU6	3.30	18	(((((((&...)))))))	1.83	5.97	1.83
3M7K	1.92	19	((((((....&.).)))))	4.06	3.89	3.89
6USO	2.54	19	(...((.........&)))	1.93	5.95	1.93
1BSU	2.00	20	((((.((((&.)))).))))	1.20	2.75	1.20
1M6G	1.65	21	((((((....&....))))))	0.64	7.49	0.64
467D	2.16	21	((((......&......))))	8.50	8.14	8.50
2HVI_1	1.98	22	(((((((((&...)))))))))	0.65	6.11	0.65
4UN4_1	2.37	22	((((((((...&..))))))))	4.27	3.54	4.27
3HOV	3.50	23	(((((............&)))))	0.59	3.39	0.59
3VH0	2.90	23	(..((((...(&)..))))...)	6.66	6.90	6.66
6KF9	3.79	24	.((((((.(.....&).)))))).	6.43	6.72	6.43
4DAV	2.20	25	....((...((.&....))...)).	10.77	4.92	4.92
2R7Z	3.80	25	((((((...........&)))))).	1.94	4.17	1.94
4FLT	2.90	25	((((((((...&.....))))))))	2.01	3.48	2.01
3L4J_2	2.48	26	((((((((((&.....))))))))))	1.72	6.08	1.72
1Y1W	4.00	27	(((((((............&)))))))	0.55	12.47	0.55
5VU8	3.20	27	..(((((((((...&...)))))))))	2.23	3.39	2.23
2HHV	1.55	28	((.(((((((.(&..).))))))).)).	3.17	4.19	3.17
4DFP	2.00	29	((((((((((((&....))))))))))))	0.49	5.57	0.49
1QSK	\	30	.(((((.....(((((.&.)))))))))).	2.68	2.72	2.68
6RAX	3.99	30	((((((((.............&))))))))	2.59	17.35	2.59
3GIM	2.70	32	((((((((((((.&......))))))))))))	1.87	5.29	1.87
6P70	1.75	41	((((((((((........&............))))))))))	1.07	4.00	1.07
3I4M	3.70	33	((((((((((...........&)))))))))).	1.60	4.31	1.60
3RB6_1	2.70	33	((((((((((((.&......)))))))))))).	0.93	4.95	0.93
2JEJ	1.86	34	(((((((((((((..&.....)))))))))))))	2.80	4.28	2.80
6H67	3.60	35	(((((((((((..........&..)))))))))))	3.68	4.05	3.68
6OW3_1	2.77	36	((((((((........&...........))))))))	1.72	4.6	1.72
2O5I_1	2.50	37	(((((((((((((..........&)))))))))))))	1.32	3.66	1.32
6M0V	3.00	37	((((((((....................&))))))))	0.44	3.48	0.44
5D4D_1	3.00	38	((((((((((......&...........))))))))))	1.7	17.72	1.7
5FJ8	3.90	39	(.(.(((((((((..........&..))))))))).)).	5.63	3.81	3.81
6OVR_1	2.84	39	((.(((((((.........&.........))))))).))	3.08	16.29	3.08
5FW1	2.50	40	((((((((....................&...))))))))	0.47	3.15	0.47
4G7H_1	2.90	41	((((((((((......&..............))))))))))	1.08	5.97	1.08
5U8S	6.10	41	((((((((((((((............&))))))))))))))	1.67	4.50	1.67
6KQE	3.30	42	((((((((((.......&..............))))))))))	1.62	6.36	1.62
1R9T	3.50	43	((((((((((((................&..))))))))))))	1.09	3.16	1.09
4QJU	2.16	43	(((..(.(((.((((.((...&.))..).))).)))).)))..	4.97	6.36	4.97
5M64	4.60	45	....(((((((.(((....&....))).)))))))..........	1.23	3.42	1.23
6JDV	3.10	47	(((((((((((........................&)))))))))))	0.74	3.11	0.74
4YG4	3.50	49	(.(((((((((((((.((((........&)))).))))))))))))).)	7.78	8.38	7.78
5T1J_2	2.95	49	(((((((((((((((((((.....&.....)))))))))))))))))))	1.92	3.61	1.92
4PU4	3.79	50	.(.((((..((((((((......(.&.)......))))))))..))))).	3.72	6.35	3.72
4C2T_1	4.00	51	((((((.(((((((((((((.....&))))))))))))).)))))).....	4.92	6.09	4.92
4NE1_3	6.50	53	((((((((((((((((((((((..(.&.)..))))))))))))))))))))))	3.02	2.13	3.02
5XOG	3.00	54	.(((((((((((((((.((((...........&)))).))))))))))))))).	2.96	3.53	2.96
5XVO_3	3.10	55	.((((((((((((((((((((((....&.))))))))))))))))))))))....	6.19	4.57	4.57
4XLP_1	4.00	56	(((((((((((((((((((((((((.....&)))))))))))))))))))))))))	0.89	4.25	0.89
2CRX	2.50	70	.(((((((((((((((((.................&.................)))))))))))))))))	1.57	6.65	1.57
6FLQ	4.10	71	((((((((((((((..(((((((((.(....&....)).))))))))..........))))))))))))))	4.80	6.04	4.80
**Average**	\	\	\	**3.0**	**5.64**	**2.83**

**Table 3 molecules-27-05936-t003:** Information and prediction accuracies (RMSD in Å) of 29 multi-chain DNAs.

PDB ID	Resolution	Length	2D Structure	Assembled	Optimized	Best
4YD2	2.47	18	(((((..&....)&))))	8.25	8.39	8.39
5VZ8	1.60	19	((((..(((&))).&))))	1.90	3.21	1.90
4YCX	2.10	20	((((..((((&))))&))))	2.18	7.78	2.18
6WIC	1.55	20	((((.(&(((&))))&))))	3.55	7.81	7.81
5D4B_2	2.66	21	.....(&(((((.&)))))).	14.15	9.60	9.60
2GWS_1	2.40	23	((((..(((((&))))).&))))	1.17	3.05	1.17
3HW8	1.95	24	((((..((((((&))))))&))))	0.75	8.04	0.75
3BAM	1.80	25	..(((((((((.&.)))&)))))).	3.30	3.92	3.30
4MKY_2	2.40	29	(((&..)))...((&(((&..))).))..	11.17	8.74	8.74
5HR9_2	2.20	31	(((.(((.(((((((&)))))))&))).)))	3.00	3.10	3.00
5J0S	2.00	33	(((((..(((((((((&))))))))).&)))))	1.07	9.61	1.07
5VRW	2.58	33	.((((.((((((((((&))))))))))&)))).	3.43	8.61	3.43
2WWY_2	2.90	35	.(.&(((((((((.(.&.).)))))))))).....	9.84	6.27	6.27
4P0Q	2.85	36	.(((((((((.(.((((&.)))).&).)))))))))	2.18	4.88	2.18
1ZR2_2	3.90	37	.(((((((((((((.((..&)).&)))))))))))))	7.50	4.43	7.50
4NCB_1	2.19	38	.((((((((((((.(.&...))&).)..))))))))).	12.60	10.63	10.63
4P0S_3	6.00	42	.((((((.(.((&)).).)))))).(..((.(&.).))..).	17.83	9.80	9.80
6L74	3.12	43	((((((((((..(...&)&..............))))))))))	2.84	6.83	6.83
4P0R_1	6.50	47	..(((((.(.((&)).).)))))....(((((((((&))))))))).	15.45	8.37	8.37
2VS8_2	2.10	53	(((((((((.(.((&(((((((((((&))))))))))))).)&.)))))))))	13.04	4.74	4.74
1TW8	2.80	55	((((((((((((.&..)))))))))).&))((((((((((.&..)))))))))).	0.90	7.95	0.90
4LD0	3.75	57	.((((((((((((((...))))))(((((((&..)))))))(((.&)))))))))))	2.00	8.42	2.00
6S16	3.41	59	(((((((((((((((...))))))(((((((((&)))))))))(((&))))))))))))	1.23	4.50	1.23
1P8K	2.60	65	((((((((((((((((((&((((((((((((.&))))))))))))))))&)))))))))))))).	11.78	2.22	11.78
4BAC	3.26	66	...((((((((((((((((&)))))))))))))))).....(((((((((((.&.)))))))))))	15.11	10.96	10.96
6X67_2	3.47	70	..(((((((((((((((((((((((&)))))))))))))))))))))))....((((((((&))))))))	1.31	11.65	1.31
1IHF	2.50	72	.((((((((((((((((((((..((((((((((((&.))))))))))))..&))))))))))))))))))))	0.41	21.0	0.41
5HOO_2	3.30	73	(((((((((((((((((((((((((&))))))))))))))))))))))))).....((((((&....))))))	18.67	12.94	12.94
2IRF	2.20	79	((((((((((((&((((((((((((&((((((((((((&.))))))))))))&))))))))))))&.))))))))))))	16.14	4.13	4.13
**Average**	\	\	\	**6.9**	**7.64**	**5.28**

**Table 4 molecules-27-05936-t004:** Information and prediction accuracies (RMSD in Å) of 24 DNAs.

PDB ID	Length	2D Structure	3dDNA			Indirect Prediction
			Assembled	Optimized	Best	
1AC7	16	((((((....))))))	2.32	5.47	2.32	3.59
1BJH	11	((((...))))	2.65	2.34	2.65	2.41
1DGO	18	(((((((....)))))))	2.84	1.97	2.84	3.49
1ECU	19	((((((((...))))))))	2.42	2.11	2.42	3.02
1EN1	18	(.(((.....))))....	8.32	3.55	3.55	6.89
1IDX	18	(((((((....)))))))	4.65	3.41	4.65	3.49
1II1	18	(((((((....)))))))	3.59	2.59	3.59	3.01
1JVE	27	((((((((((((...))))))))))))	0.76	1.92	0.76	4.90
1KR8	7	((...))	2.33	1.22	2.33	1.99
1LA8	13	(((((...)))))	1.75	3.15	1.75	3.98
1NGO	27	(((((((((((.....)))))))))))	3.03	3.87	3.03	6.42
1NGU	27	((((..(((((.....)))))..))))	3.93	6.58	3.93	8.59
1P0U	13	(((((...)))))	1.07	1.79	1.07	3.28
1PQT	7	((...))	2.49	1.24	2.49	1.87
1QE7	18	(((((((....)))))))	3.43	2.17	3.43	3.47
1XUE	17	((...((...))...))	0.95	3.55	0.95	4.19
1ZHU	10	..((...)).	3.15	6.88	3.15	3.05
2K71	8	((....))	3.89	1.68	3.89	2.56
2L5K	23	(((...((((...))))...)))	3.47	3.2	3.47	4.07
2LO5	12	((((...).)))	2.67	1.74	2.67	4.01
2LO8	10	(((...).))	2.80	2.33	2.80	5.53
2M8Y	15	((((((...))))))	0.83	3.11	0.83	3.56
2VAH	18	(((((((....)))))))	0.45	1.39	0.45	4.10
2VAI	18	(((((((....)))))))	0.47	1.39	0.47	4.07
**Average**	\	\	**2.67**	**2.86**	**2.47**	**4.00**

## Data Availability

The web server to predict DNA 3D structures is available at hust.edu.cn/new/3dRNA, and the validation data can also be downloaded at the web server.

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
