# Peer review of "3dDNA: A Computational Method of Building DNA 3D Structures"

_molecules, 2022, doi:10.3390/molecules27185936_

Round 1
Reviewer 1 Report
Zhang et al. introduces 3dDNA which is analogous to 3dRNA also they developed for ssDNA structure prediction. Need for this tool is clear, and the paper is well-laid out with proper sets of examples.
My only complaint is that the half of conclusion part is listing its limitation. Can it be rewritten in more positive tone?
Author Response
Reply to the reviewer1
We thank you very much for your valuable suggestions and we have revised our manuscript accordingly.
Review 1
Zhang et al. introduces 3dDNA which is analogous to 3dRNA also they developed for ssDNA structure prediction. Need for this tool is clear, and the paper is well-laid out with proper sets of examples.
My only complaint is that the half of conclusion part is listing its limitation. Can it be rewritten in more positive tone?
Answer: Thanks for your nice suggestions, we have rewritten the conclusion.
Reviewer 2 Report
The authors present a method using template matching to predict 3D structures of folded DNAs. Their methods closely follow those which predict RNA structures such as iFoldRNA and RNA composer. Previous attempts to predict DNA structures have been limited primarily to DNA hairpins, whereas the present study is expanded to include multi-strand structures. Further, the authors report improved accuracy of the structure predictions. While this study represents a step forward which will be beneficial to many, the work is not presented clearly and in a manner that appeals to a more broad audience. I have many concerns, both major and minor; listed below.
Major concerns:
1. The introduction is not tailored to a non-expert audience. It is only mentioned in passing (last line of first introduction paragraph) that similar template-based approaches have been quite successful at predicting RNA structures over the past decade. Because those works serve as the foundation for this study, they should be described in more detail. Specifically, how do they differ from one another? How do the differences translate into better/worse predictions? Did these aspects influence decisions you made regarding your current approach for DNA?
2. The Methods section lacks details that are important to understand how this new tool works. The explanation of the decomposition was well-done, but the rest is not sufficient. Some suggested details to include would be: (a) When templates are being matched, does this tool require exact sequence matching in non-basepaired segments? (B) How are helices handled? Do they simply conform to standard B-form structure or do they include sequence-induced bending? (C) In Figure 1, the break loop and open loop appear the same, why are they handled differently? (D) It is noted that "clean", "mutate", and "amber" are used. While the reference is good for a detailed description, it is important to give a less technical, brief description of what those procedures do in your method. (E) The description of SSE scoring is insufficient. What defines a family? What does it mean for a sequence to be traversed? What score is required for an SSE to be "Found" in the 3dDNA_Lib? (F) What type of minimization is performed using the AMBER 98 force field? Steepest descent? Conjugate gradient? A combination of the two? (G) The 3dRNAscore should be cited and briefly described. (H) How is it determined whether a structure needs to be futher optimized using SAMC? In other words, what is required for a structure to have a "perfect template"?
3. Five out of the 24 references are self-references. Considering the abundance of literature regarding nucleic acid structure prediction, this is a concerning ratio.
4. The conclusions are sparse. Please put your findings in context with the broader literature. How do the RMSDs compare to those of other DNA prediction methods? How do they compare to RNA prediction methods? Other than lack of experimental DNA structures, are there any inherent properties of DNA/RNA that would make template-based predictions better or worse? I would suspect that steric clashes caused by the 2'OH in RNA would limit conformational space, making DNA structure more difficult to predict.
Minor concerns:
1. Please change the colors in the figures to accommodate red-green color-blindness.
2. On page 11, please describe the "indirect" method and how it differs from the method presented.
3. The use of ssDNA throughout the manuscript gets confusing, especially when describing multi-chain DNAs. Rather than calling them ssDNA, it would be simpler to just call them DNA and note that your software excludes those DNAs that are double-stranded and perfectly complementary.
4. It would be helpful to have secondary structure structure depictions included in Figures 5, 7 and 9 (similar to what is shown in Figure 4).
5. In the tables, it would be helpful to include the reported resolution of the structures in the PDB.
6. In Table 1, the RMSD for 1EZN is quite high despite the fact that it is mostly basepaired. One would expect this structure to be predicted well. Why do you suspect this structure is so poor?
7. In Figure 4, it would be easier to see if the structures in B and C were in the same orientation.
8. The manuscript requires extensive editing for clarity and grammar. Examples listed below:
-Last sentence of introduction, please add a comma between "following" and "DNA" and change the word "no" to "not".
-Sixth line of page 2, please remove the word "the" from between "that" and "any".
-Tenth line of page 2, please add "a" between "in" and "break".
-Figure 1 caption, please change from "RNA 2D plots" to "DNA 2D plots".
-Please fix remaining grammatical mistakes.
Author Response
Reply to the reviewer2
We thank you very much for your valuable comments and suggestions and we have revised our manuscript accordingly. The following are our replies.
The authors present a method using template matching to predict 3D structures of folded DNAs. Their methods closely follow those which predict RNA structures such as iFoldRNA and RNA composer. Previous attempts to predict DNA structures have been limited primarily to DNA hairpins, whereas the present study is expanded to include multi-strand structures. Further, the authors report improved accuracy of the structure predictions. While this study represents a step forward which will be beneficial to many, the work is not presented clearly and in a manner that appeals to a more broad audience. I have many concerns, both major and minor; listed below.
Major concerns:
- The introduction is not tailored to a non-expert audience. It is only mentioned in passing (last line of first introduction paragraph) that similar template-based approaches have been quite successful at predicting RNA structures over the past decade. Because those works serve as the foundation for this study, they should be described in more detail. Specifically, how do they differ from one another? How do the differences translate into better/worse predictions? Did these aspects influence decisions you made regarding your current approach for DNA?
Answer: The introduction has been revised according to your valuable suggestions, mainly including introduction of two main approaches of RNA 3D structure prediction and giving a brief explaining why we use template-based approach.
- The Methods section lacks details that are important to understand how this new tool works. The explanation of the decomposition was well-done, but the rest is not sufficient. Some suggested details to include would be: (a) When templates are being matched, does this tool require exact sequence matching in non-basepaired segments? (B) How are helices handled? Do they simply conform to standard B-form structure or do they include sequence-induced bending? (C) In Figure 1, the break loop and open loop appear the same, why are they handled differently? (D) It is noted that "clean", "mutate", and "amber" are used. While the reference is good for a detailed description, it is important to give a less technical, brief description of what those procedures do in your method. (E) The description of SSE scoring is insufficient. What defines a family? What does it mean for a sequence to be traversed? What score is required for an SSE to be "Found" in the 3dDNA_Lib? (F) What type of minimization is performed using the AMBER 98 force field? Steepest descent? Conjugate gradient? A combination of the two? (G) The 3dRNAscore should be cited and briefly described. (H) How is it determined whether a structure needs to be futher optimized using SAMC? In other words, what is required for a structure to have a "perfect template"?
Answer: The Method section has been revised according to your valuable suggestions and more details have been given.
- When matching templates, secondary structure, loop sequence information will be taken into account at the same time.
- In the template library, we use the helix of native DNA as a template, so our helix template includes sequence-induced bending due to the fact that not all native DNA helices are standard B-helices.
- Although the open loop and broken loop in Figure 2 look the same, their secondary structures are different. The secondary structures of the open loop and broken loop are “.(()).” and “((. &.))”, can lead to significant differences in both search templates and module assembly. And there's only one open loop in DNA, but there can be multiple open loops depending on the number of strands. The definition of broken loop and open loop in original manuscript was indeed ambiguous, so we have redefined them in revised manuscript.
- “clean”, “muate”, and “amber” have been briefly described in revised manuscript.
- In the process of selecting the most suitable template for each SSE, we mainly based on two criteria: secondary structure and sequence. First, compare the secondary structures of SSE and search template. If the secondary structures match, then add 5 points. Secondly, the sequence of SSE and the template to be searched was compared. For each nucleotide in the loop region, if it is the same, the residue will be scored 1, while for each nucleotide in the helix region, if it is the same, the residue will be scored 0.2.
- “Steepest descent” is performed using the AMBER 98 force field.
- The 3dRNAscore has been cited and briefly described.
- “Perfect template" was not clearly explained and has been redefined.
- Five out of the 24 references are self-references. Considering the abundance of literature regarding nucleic acid structure prediction, this is a concerning ratio.
Answer: 3dDNA is expanded from our RNA 3D structure prediction method 3dRNA and this is why we quote our articles. According to your suggestion, we revised the introduction again, so we have cited more articles about the prediction of RNA 3D structure.
- The conclusions are sparse. Please put your findings in context with the broader literature. How do the RMSDs compare to those of other DNA prediction methods? How do they compare to RNA prediction methods? Other than lack of experimental DNA structures, are there any inherent properties of DNA/RNA that would make template-based predictions better or worse? I would suspect that steric clashes caused by the 2'OH in RNA would limit conformational space, making DNA structure more difficult to predict.
Answer:Thanks for these good suggestions, we have revised the inclusion. Since no other prediction methods of DNA 3D structures, we can only compare 3dDNA with the indirect method proposed by Jeddi group.
Minor concerns:
- Please change the colors in the figures to accommodate red-green color-blindness.
Answer: We replaced the red color with another color in related Figures.
- On page 11, please describe the "indirect" method and how it differs from the method presented.
Answer: We briefly introduced the only two indirect DNA prediction methods. The “indirect” method predicted the 3D structure of the corresponding RNA, then converted it into that of DNA by replacing the nucleotide U with T, and finally refined the resulting 3D structures through energy minimization. our method 3dDNA is the first fully automated method proposed to predict DNA tertiary structure from DNA sequences.
3.The use of ssDNA throughout the manuscript gets confusing, especially when describing multi-chain DNAs. Rather than calling them ssDNA, it would be simpler to just call them DNA and note that your software excludes those DNAs that are double-stranded and perfectly complementary.
Answer: We added the classification of DNA structures in PDB and explained the DNA types predicted by 3dDNA.
- It would be helpful to have secondary structure depictions included in Figures 5, 7 and 9 (similar to what is shown in Figure 4).
Answer: We added the secondary structures of DNAs to the figure in the corresponding figures.
- In the tables, it would be helpful to include the reported resolution of the structures in the PDB.
Answer:The resolutions are added if available.
- In Table 1, the RMSD for 1EZN is quite high despite the fact that it is mostly basepaired. One would expect this structure to be predicted well. Why do you suspect this structure is so poor?
Answer: The structure of 1EN1 is special, including a three-way junction. There is no matching template structure in the template library, so the predicted structure of 1EN1 is bad.
- In Figure 4, it would be easier to see if the structures in B and C were in the same orientation.
Answer: We redraw Figure 4 to satisfy that the structure directions of model1 in B and C are exactly the same.
- The manuscript requires extensive editing for clarity and grammar. Examples listed below:
-Last sentence of introduction, please add a comma between "following" and "DNA" and change the word "no" to "not".
-Sixth line of page 2, please remove the word "the" from between "that" and "any".
-Tenth line of page 2, please add "a" between "in" and "break".
-Figure 1 caption, please change from "RNA 2D plots" to "DNA 2D plots".
-Please fix remaining grammatical mistakes.
Answer: We have checked the full text again and corrected the grammatical errors
Reviewer 3 Report
For “3dDNA: a computational method of building single-stranded DNA 3D structures” by Zhang et al.
The authors proposed a method for predicting the three-dimensional structures of DNAs based on fragment assembly, in a comparative way to a well-established method of 3dRNA for RNAs. By searching the appropriate fragment in DNA and RNA fragment libraries, optimizing and clustering process, the three-dimensional structures of DNAs can be generated. Importantly, the authors provide an online-webserver of the method for users. This is really a very important work. I am very pleased to recommend its acceptance. However, I still have the following very minor issues since I am really interested in the proposed method.
1, The work is not only very helpful for readers in the field of DNA structures, but also can be very referential for readers in modeling RNA structures. Moreover, the proposed method was developed according to 3dRNA for modeling RNA 3D structures and also involves the templates of RNA fragments when those of DNAs are absent.
Thus, the authors may expand the introduction for reviewing the progress in RNA 3D structure prediction, in addition to that in DNA structure modeling. Such expanded introduction will be very helpful for readers in the field of both DNA and RNA.
2, The authors reserved 2-base-pairs at the end of each fragment when building the library. If a fragment is less than 2bp in the end, how do the authors deal with the case when building the whole structure? For example, 6l74 in Table 3 which has a break loop ‘(... &)’.
3, During the assembly process, the authors matched an optimal fragment for each SSE through certain rules. Would better structures be generated if more fragments were selected? Moreover, by testing the web version and only assemble procedure, why multiple structures are assembled when there is only one fragment of each SSE?
4, The author divides the helix that connects broken single chains into open loop and break loop. Is there any specific difference in the building process, or only bracket-dot notations of the secondary structures are different? When the author predicts the double-chain and multi-chain structures, the prediction accuracy of DNAs with double-chains and multi-chains is lower than that of the single chain, is this related to the existence of break loop in the double-chains and multi-chains structures? I am very curious about that.
Author Response
Reply to the reviewer3
We thank you very much for your valuable comments and suggestions and we have revised our manuscript accordingly. The following are our replies.
For “3dDNA: a computational method of building single-stranded DNA 3D structures” by Zhang et al.
The authors proposed a method for predicting the three-dimensional structures of DNAs based on fragment assembly, in a comparative way to a well-established method of 3dRNA for RNAs. By searching the appropriate fragment in DNA and RNA fragment libraries, optimizing and clustering process, the three-dimensional structures of DNAs can be generated. Importantly, the authors provide an online-webserver of the method for users. This is really a very important work. I am very pleased to recommend its acceptance. However, I still have the following very minor issues since I am really interested in the proposed method.
1, The work is not only very helpful for readers in the field of DNA structures, but also can be very referential for readers in modeling RNA structures. Moreover, the proposed method was developed according to 3dRNA for modeling RNA 3D structures and also involves the templates of RNA fragments when those of DNAs are absent.
Thus, the authors may expand the introduction for reviewing the progress in RNA 3D structure prediction, in addition to that in DNA structure modeling. Such expanded introduction will be very helpful for readers in the field of both DNA and RNA.
Answer: The introduction of this paper is indeed quite concise, we described the progress of RNA and DNA prediction in the revised version.
2, The authors reserved 2-base-pairs at the end of each fragment when building the library. If a fragment is less than 2bp in the end, how do the authors deal with the case when building the whole structure? For example, 6l74 in Table 3 which has a break loop ‘(... &)’.
Answer: According to our SSE decomposition procedure, the base pairing you refer to should be included in this SSE with secondary structure ((... (&) & ..............)), instead of the SSE with secondary structure "(...&)" that you proposed, so there will not be a problem.
3, During the assembly process, the authors matched an optimal fragment for each SSE through certain rules. Would better structures be generated if more fragments were selected? Moreover, by testing the web version and only assemble procedure, why multiple structures are assembled when there is only one fragment of each SSE?
Answer: We just set a set of rules to score the templates of each SSE, instead of just picking the best template for each SSE. In fact, our assembly will combine all the templates searched by SSE to get a large number of assembled structures. The web server sets the default to be five.
4, The author divides the helix that connects broken single chains into open loop and break loop. Is there any specific difference in the building process, or only bracket-dot notations of the secondary structures are different? When the author predicts the double-chain and multi-chain structures, the prediction accuracy of DNAs with double-chains and multi-chains is lower than that of the single chain, is this related to the existence of break loop in the double-chains and multi-chains structures? I am very curious about that.
Answer:
- The secondary structures of the open loop and broken loop are “.(()).” and “((. &.))”, can lead to significant differences in both search templates and module assembly. And there's only one open loop in DNA, but there can be multiple open loops depending on the number of chains.
- There are a large number of broken loop structures in double-chain and multi-chain RNAs, especially in the latter, but such broken loop structures are very uncommon in template library, that is, perfect templates are basically not found in template library, which leads to their prediction accuracy worse than that of single chain.